# High-Intensity Exercise Training Improves Basal Platelet Prostacyclin Sensitivity and Potentiates the Response to Dual Anti-Platelet Therapy in Postmenopausal Women

**DOI:** 10.3390/biom12101501

**Published:** 2022-10-17

**Authors:** Kate A. Wickham, Line B. Nørregaard, Martina H. Lundberg Slingsby, Stephen S. Cheung, Ylva Hellsten

**Affiliations:** 1August Krogh Section for Human Physiology, Department of Nutrition, Exercise and Sports, University of Copenhagen, 2100 Copenhagen, Denmark; 2Environmental Ergonomics Lab, Department of Kinesiology, Brock University, St. Catharines, ON L2S 3A1, Canada

**Keywords:** dual anti-platelet therapy, exercise, menopause, platelets, coagulation, thrombosis, prostacyclin

## Abstract

The risk of thrombotic events dramatically increases with age and may be accelerated in women by the cessation of endogenous estrogen production at menopause. Patients at risk of thrombosis are prescribed dual anti-platelet therapy (DAPT; aspirin and a P2Y_12_ antagonist) and are encouraged to participate in regular physical activity, as these modalities improve nitric oxide and prostacyclin-mediated inhibition of platelet aggregation. Methods: We assessed prostacyclin sensitivity as well as basal platelet reactivity with and without in vitro DAPT before and after an 8-week high-intensity exercise training program in 13 healthy, sedentary postmenopausal women. The training intervention consisted of three 1 h sessions per week. Isolated platelets were analyzed for thromboxane A_2_ receptor, thromboxane A_2_ synthase, cyclooxygenase-1, and prostacyclin receptor protein expression. Additionally, plasma 6-keto prostaglandin F_1__α_ and thromboxane B_2_ levels were determined. Results: Exercise training made platelets more sensitive to the inhibitory effects of prostacyclin on thromboxane-, collagen-, and adenosine 5′-diphosphate (ADP)-induced aggregation, as well as thrombin-receptor activator peptide 6- and ADP-induced aggregation with DAPT. However, there was no change in protein expression from isolated platelets or plasma thromboxane B_2_ and prostacyclin levels following training. Conclusion: Together, these findings emphasize the importance of promoting physical activity as a tool for reducing thrombotic risk in postmenopausal women and suggest that training status should be considered when prescribing DAPT in this cohort.

## 1. Introduction

Thrombotic events are a leading cause of death, affecting one in four people worldwide [1], and the risk of thrombosis dramatically increases after the age of 60 [2]. In healthy individuals, circulating platelets are activated in response to vascular injury to promote platelet aggregation and subsequent thrombus formation [3]. However, aging is associated with an increase in circulating factors that promote platelet activation without a proportional increase in factors that prevent platelet activation, contributing to increased thrombotic risk [4,5,6]. In a large cohort study of men and women, each successive decade of life was linked to an ~8% increase in platelet aggregability [7]. Notably, compared with men, women may have an exaggerated increase in platelet hyperreactivity and thrombotic risk with age due to menopause, and the concomitant loss of endogenous estrogen [8,9,10]. Estrogen inhibits platelet reactivity by stimulating the production of key endothelial-derived platelet inhibitors including nitric oxide (NO) [11] and prostacyclin [12]. Therefore, in menopause, when estrogen levels are very low, the balance is shifted in favour of platelet activation over inactivation, and thus blood clot formation, which contributes to the elevated risk of thrombosis in this cohort [8,10]. 

Patients who have experienced a thrombotic event, or who are deemed at risk of thrombosis, are typically prescribed dual anti-platelet therapy (DAPT) for 12 months [13] to reduce platelet hyperreactivity and the formation of harmful blood clots. DAPT also dramatically increases the anti-platelet effects of NO and prostacyclin [14,15,16]. DAPT consists of aspirin, which inhibits platelet production of thromboxane A_2_ and a P2Y_12_ receptor antagonist (e.g., ticagrelor, prasugrel, or clopidogrel), which inhibits the activation of platelet adenosine 5′-diphosphate (ADP) P2Y_12_ receptors. In addition to DAPT, the patients are encouraged to participate in regular physical activity, due to its established array of beneficial effects on vascular health [13,17,18]. Importantly, in postmenopausal women, regular exercise mimics many of the same cardioprotective effects as endogenous estrogen, and the reductions in platelet hyperreactivity are at least partially attributed to exercise-induced improvements in NO and prostacyclin bioavailability [19,20,21] and platelet prostacyclin sensitivity [9]. Taken together, regular exercise and DAPT both elicit anti-platelet effects via NO and prostacyclin-mediated pathways, highlighting the potential for a synergistic effect. 

Previous research from our laboratory has shown that well-trained middle-aged men are more sensitive to DAPT (aspirin and ticagrelor) than age-matched untrained individuals [22]. This demonstrates that in men, training status should be considered when prescribing DAPT. However, given that current health guidelines recommend concomitant DAPT and regular physical activity interventions [13], which both provide potent anti-platelet effects, it is critical to evaluate the direct impact of an exercise training program on the efficacy of DAPT in previously sedentary individuals. Moreover, several reviews have recently drawn attention to the severe underrepresentation of women in DAPT research initiatives [23,24], as well as preliminary evidence for lower DAPT efficacy in women compared with men [25,26]. Therefore, the purpose of this study is to assess the impact of an 8-week high-intensity exercise training program on (1) prostacyclin sensitivity with and without in vitro DAPT, (2) basal platelet reactivity with and without in vitro DAPT; (3) isolated platelet protein expression, and (4) circulating levels of platelet activators and inhibitors in postmenopausal women. We hypothesized that 8 weeks of high-intensity exercise training would improve platelet prostacyclin sensitivity with and without in vitro DAPT, which would be mediated by increases in platelet prostacyclin receptor expression and circulating prostacyclin.

## 2. Materials and Methods

The experimental protocol was approved by the ethics committee of Copenhagen (H-20037633). All participants provided written informed consent and experiments were conducted in accordance with the Declaration of Helsinki.

### 2.1. Recruitment of Subjects

Sixteen participants were recruited for an 8-week exercise training intervention via an online participant database and newspaper advertisements. The inclusion criteria were sedentary (no regular physical activity in >2 years) but otherwise healthy, postmenopausal women (>12 months since last menstrual bleeding, postmenopausal status was confirmed by blood hormone levels; Table 1), 50 to 70 years old, body mass index (BMI) < 30 kg∙m^−2^, and normotensive (blood pressure < 130/90 mmHg). Participants were excluded if they smoked in the last 10 years or had excessive alcohol intake (>14 units per week as well as blood alanine transferase and aspartate aminotransferase levels). Other exclusion criteria were the use of regular medication, hormone replacement therapy, or phytoestrogen supplements (e.g., soybean or red clover products).

### 2.2. Study Design

The study design consisted of a health screening day, two body composition days (pre- and post-training), two experimental days (pre- and post-training), and 8 weeks of high-intensity exercise training. For all laboratory visits, participants were instructed to avoid caffeine for 24 h, strenuous exercise for 48 h, and non-steroidal anti-inflammatory drugs for at least 7 days prior to participation.

### 2.3. Health Screening Day

All participants underwent an examination to ascertain their health status, eligibility to participate in exercise training, and that all inclusion criteria were met. The health examination included: a 10-point resting electrocardiogram (ECG), blood samples for hematological markers, sex hormones, and cholesterol (analyzed within 2 h at Rigshospitalet in Copenhagen, Denmark) (Table 1 and Table 2). Participants also completed the short version of the International Physical Activity Questionnaire (IPAQ) to quantify their self-reported physical activity habits.

### 2.4. Exercise Training

The participants underwent 8 weeks of high-intensity exercise training, which involved 1 h of training performed 3 times per week. The training program was accepted as completed when participants performed a minimum of 20 sessions, and training was terminated at a maximum of 26 sessions over the 8-week period. The training included a combination of small-sided floorball matches and interval-based cycle training (spinning). This mixed modality, high-intensity training program was selected to maintain participant motivation and avoid overuse injuries. Participants were required to participate in at least one session of floorball and one session of spinning per week with the option to select the third session based on preference. Each training session was monitored by members of the research team and encouragement was provided to facilitate high-intensity exercise. Additionally, heart rate (HR) was monitored continuously (POLAR TEAM Pro, Polar Electro Oy, Kempele, Finland) throughout the exercise sessions to quantitatively ensure a high exercise intensity was maintained (average HR of the entire session > 75% HR_max_). HR data were quantified using time spent in HR zones, where Zone 1 was ≤60% HR_max_, Zone 2 was 61 to 70% HR_max_, Zone 3 was 71 to 80% HR_max_, Zone 4 was 81 to 90% HR_max_, Zone 5 was ≥91% HR_max_, as defined by POLAR TEAM Pro (Polar Electro Oy, Kempele, Finland).

### 2.5. Body Composition Days

Before and after the training intervention, participants arrived at the laboratory in a fasted state and rested in a supine position for 15 min before a dual-energy X-ray absorptiometry (DXA) scan was performed to assess body composition.

### 2.6. Experimental Days

Experimental days were performed pre- and post-training. The participants arrived at the laboratory in a semi-fasted state (≥2 h since last meal) and were instructed to repeat their diet for the post-training experimental day. A diet record was utilized to ensure successful repetition. Then, resting blood samples were drawn into 3.2% sodium-citrate tubes (454332, Greiner Bio-One, Frickenhausen, Germany) for basal and prostacyclin sensitivity platelet reactivity assays (~34 mL) and CTAD tubes (~18 mL; 454064, Greiner Bio-One, Frickenhausen, Germany) for platelet isolation. Participants then performed a maximal oxygen uptake (V̇O_2max_) test on a cycle ergometer for determination of fitness status. The test consisted of a 5 min warm-up at 50 Watts (W) followed by 25 W∙min^−1^ increases until volitional exhaustion. Participants then rested for 10 min before completing a V̇O_2max_ verification at 110% W_max_ until volitional exhaustion.

### 2.7. Basal Platelet Reactivity Assay

The primary outcomes of this assay were to: (1) determine the effects of exercise training on platelet aggregation and (2) evaluate the anti-aggregatory effects of DAPT pre- and post-training. To perform this assay, 18 mL of blood was drawn into 3.2% sodium–citrate tubes. The blood was immediately analyzed for mean platelet count, red blood cells (RBC), and white blood cells (WBC) using a hematology analyzer (Sysmex XN-2100, Sysmex, Kōbe, Japan). Blood samples were then centrifuged at 180 g for 10 min at 20 °C to obtain platelet-rich plasma (PRP). The PRP was pipetted into two 2 mL aliquots for treatment with vehicle or DAPT. The remaining blood was centrifuged further at 15,000 g for 2 min at 20 °C to obtain platelet-poor plasma (PPP). One 2 mL aliquot of PRP was treated with a vehicle consisting of 0.5% DMSO and 0.03% ethanol in 1X phosphate-buffered saline (PBS, Gibco PBS (10X), pH 7.2, Fisher Scientific, Waltham, MA, USA). The second 2 mL aliquot of PRP was treated with DAPT consisting of the reversible purinergic P2Y_12_ receptor antagonist Ticagrelor (1 µM; 15425, Cayman Chemical Company, Ann Arbor, MI, USA) and aspirin (acetylsalicylic acid, 100 µM; 5376, Sigma-Aldrich, St. Louis, MO, USA) [22,27] at concentrations that reflect in vivo platelet reactivity in the presence of DAPT [28,29]. The PRP aliquots were incubated for 30 min at 37 °C in a water bath. The PRP and PPP were then pipetted into a 96-well round bottom plate (650101, Greiner Bio-One, Frickenhausen, Germany) and, as per our previous studies [9,22], 40 µL was subsequently added via multi-channel pipette to a 96-well half-area plate (675161, Greiner Bio-One, Frickenhausen, Germany) pre-coated with known concentrations of platelet agonists: collagen (0.0156–16 µg∙mL^−1^; 1130630, Takeda, Tokyo, Japan), thrombin receptor activator peptide 6 (0.11–40 µM; TRAP-6, 4017752, Bachem, Bubendorf, Switzerland), epinephrine (0.001–10 µM; E4375, Sigma-Aldrich, St, Louis, MO, USA), ADP (0.08–80 µM; A2754, Sigma-Aldrich, St. Louis, MO, USA), and thromboxane A_2_ mimetic U46619 (0.02–40 µM; 16450, Cayman Chemical Company, Ann Arbor, MI, USA). The 96-well half-area plate was covered with Parafilm and put on an orbital plate shaker (Thermomixer C, Eppendorf, Hamburg, Germany) for 5 min at 37 °C and 1200 rpm to allow aggregation to occur. The 96-well half-area plate was immediately analyzed using a plate reader (Emax, Molecular Devices, San Jose, CA, USA), where absorbance was measured at 595 nm and platelet aggregation (%) was calculated:$$Platelet\;aggregation\;\left(\%\right)=1-\left(\frac{sample-PPP}{PRP-PPP}\right)\times 100\%$$

### 2.8. Prostacyclin Sensitivity Assay

The primary outcomes of this assay were to: (1) determine the effects of exercise training on the anti-aggregatory action of prostacyclin and (2) evaluate the combined anti-aggregatory effects of prostacyclin and DAPT pre- and post-training. To perform this assay, 16 mL of blood drawn into 3.2% sodium-citrate tubes was used. The centrifugation steps and treatment with vehicle or DAPT were the same as the basal platelet reactivity assay. The PRP was then pipetted into aliquots on a 96-well round bottom plate (650101, Greiner Bio-One, Frickenhausen, Germany) and, to mimic a physiological milieu, was treated with low concentrations of prostacyclin (1–300 nM; epoprostenol 2989, R&D Systems, Minneapolis, MN, USA) [22] for 1 min at room temperature before 40 µL of this treated PRP was added via multi-channel pipette to a 96-well half-area plate (675161, Greiner Bio-One, Frickenhausen, Germany) pre-coated with known concentrations of platelet agonists: collagen (10 µg∙mL^−1^), TRAP-6 (10 µM), ADP (20 µM), and U46619 (10 µM). The 96-well half-area plate was covered with Parafilm and underwent the same shaking and absorbance measurement as previously described. All assays were completed within 2 h of drawing the blood.

### 2.9. Platelet Isolation

Blood drawn into CTAD tubes (18 mL) was centrifuged for 10 min at 180 *g* and 20 °C to obtain PRP, which was transferred to a 15 mL tube and then treated with 1 µL prostaglandin E_1_ (P5515, Sigma-Aldrich, St. Louis, MO, USA) per 100 µL PRP to reversibly inhibit platelet activation. The PRP was centrifuged for 5 min at 5000× *g* and 20 °C to form a platelet pellet. The supernatant was removed and platelet wash buffer (140 mM NaCl, 5 mM KCl, 12 mM sodium citrate, 10 mM glucose, 12.5 mM sucrose, pH 6.0) was added in equal volumes to the initial volume of PRP. The pellet and wash buffer were centrifuged for 5 min at 5000× *g* and 20 °C. The supernatant was removed, and the platelet pellet was resuspended in 400 µL of platelet resuspension buffer (10 mM HEPES, 140 mM NaCl, 3 mM KCl, 0.5 mM MgCl_2_, 5 mM NaHCO_3_, 10 mM glucose, pH 7.4). The platelet count was obtained (Sysmex XN-2100, Sysmex, Kobe, Japan), then an equal volume of 0.5% Triton X-100 (T8787, Sigma-Aldrich, St. Louis, MO, USA) in 1X PBS (Gibco PBS (10X), pH 7.2, Fisher Scientific, Waltham, MA, USA) was added to the resuspended platelets to ensure lysis prior to storage at −80 °C. Platelet isolation was initiated within 3 hours of drawing the blood.

### 2.10. Western Blotting of Isolated Platelets

Platelet cell lysates were prepared for Western blot analysis by adding concentrated sample buffer (0.5 M Tris-base, DTT, SDS, glycerol, and bromophenol blue) and heated for 3 min at 96 °C prior to loading. Approximately 1.9 × 10^6^ platelets were loaded per well using pre-cast Criterion TGX stain-free gels (4–15%) (Bio-Rad, Hercules, CA, USA). Stain-free (TGX) images of the gels were obtained as a loading control. Then, proteins were semi-dry transferred to a polyvinylidene difluoride membrane (Immobilon Transfer Membrane, Millipore, Burlington, MA, USA). The membranes were incubated overnight at 4 °C with primary antibodies diluted in either 5% milk (thromboxane A_2_ receptor; ab233288, 1:1000 and cyclooxygenase-1; ab133319, 1:2000) or 3% BSA (thromboxane A_2_ synthase; ab39362, 1:1000 and prostacyclin receptor; ab196653, 1:1000). Next, the membranes were washed for 5 min with TBST before adding anti-rabbit secondary horseradish peroxidase-conjugated antibody (111-035-144, Jackson Immunoresearch, West Grove, PA, USA) for 1 h. Bands were visualized with Luminata Forte (Merck Millipore, Burlington, MA, USA). The images were digitized on a ChemiDoc MP system (Bio-Rad, Hercules, CA, USA). All proteins were expressed in arbitrary units normalized to the average of all samples loaded on the gel.

### 2.11. Plasma Thromboxane B_2_ and 6-Keto Prostaglandin F_1__α_ Assays

Three mL of blood was collected in an EDTA-coated tube and was immediately centrifuged at 4000 rpm for 5 min at 5 °C. Plasma was stored at −80 °C until future analysis. Plasma concentrations of prostacyclin and thromboxane A_2_ were determined via their breakdown products, 6-keto prostaglandin F_1α_ (6-keto PGF_1α_) and thromboxane B_2_, respectively. The Thromboxane B_2_ (KGE011, R&D Systems, Minneapolis, MN, USA) and 6-keto PGF_1α_ (515211, Cayman Chemical Company, Ann Arbor, MI, USA) assays were performed according to the manufacturer’s instructions.

### 2.12. Statistical Analyses

The sample size of 15 participants was determined by performing a power calculation with an α level of 0.05 and a power level of 0.8 for a 50% increase in platelet sensitivity to the inhibitory effects of prostacyclin, based on previous reports [9,22,30]. We recruited 16 participants, and the final data set includes 13 participants due to 3 dropouts (COVID-19, injury, and inability to commit to the training program). Statistical analyses were performed using R-studio (Version 4.1.2, R Foundation for Statistical Computing, Vienna, Austria). Figures were created using GraphPad Prism (GraphPad Software, Version 9.3.1, San Diego, CA, USA). Basal platelet reactivity and prostacyclin sensitivity were performed using linear mixed models with a Tukey post hoc test. The half maximal effective concentration (EC_50_) was determined as the concentration of agonist required to elicit half of the maximal platelet aggregation. The half maximal inhibitory concentration (IC_50_) was determined as the concentration of prostacyclin required to elicit half of the maximal inhibition of platelet aggregation. The effects of exercise training on health parameters, EC_50_, IC_50_, platelet Western blots, and ELISA parameters were analyzed using a paired two-tailed t-test. The normality of the data was confirmed using Q-Q plots. Statistical significance was accepted at *p* ≤ 0.05. Data are reported as mean ± standard deviation (SD). The final number of replicates is indicated in each figure legend.

## 3. Results

### 3.1. Participant Characteristics

Thirteen healthy, but sedentary, postmenopausal women completed the 8-week high-intensity training intervention. A summary of the participant characteristics can be found in Table 1.

There was no statistically significant increase in absolute (mL O_2_∙min^−1^; *p* = 0.15) or relative (mL O_2_∙kg^−1^∙min^−1^; *p* = 0.14) V̇O_2max_. However, after the training period, the participants showed a significant increase in lean body mass (LBM; *p* = 0.01), in addition to a significant reduction in relative fat mass (*p* = 0.02) and an increase in relative lean mass (*p* = 0.02). An overview of the body composition, hematological, as well as health and fitness parameters pre- and post-training is provided in Table 2.

### 3.2. Exercise Training

The participants completed a total of 23 ± 2 training sessions over the 8-week period, of which 9 ± 3 sessions were spinning, and 13 ± 3 sessions were floorball training. The average training session was 53 min 56 sec ± 1 min 5 sec. Time spent in the different HR zones over the exercise sessions is depicted in Figure 1.

### 3.3. Exercise Training and Basal Platelet Reactivity

Basal platelet reactivity was evaluated as the aggregation (%) to a known concentration of platelet agonist as well as the concentration of agonist required to elicit half of the maximal platelet aggregation (EC_50_). There was a main effect of concentration, whereby increasing concentrations of collagen, TRAP6, ADP, U46619, and epinephrine resulted in higher levels of platelet aggregation (*p* < 0.0001 for all agonists). The exercise intervention increased platelet aggregation by ~28% to U46619 (1.98 µM), a thromboxane receptor analogue (65.8 ± 24.9 vs. 84.5 ± 7.6% aggregation; *p* = 0.0002) (Figure 2C,D) (lower EC_50_; *p* = 0.04; Figure 3E). However, the training intervention did not affect platelet aggregation or the EC_50_ to collagen, TRAP6, ADP, or epinephrine (Figure 2 and Figure 3).

### 3.4. Exercise Training, Dual Anti-Platelet Therapy, and Basal Platelet Reactivity

There was a main effect of condition, whereby DAPT significantly inhibited collagen-, TRAP6-, ADP-, U46619-, and epinephrine-induced platelet aggregation (*p* < 0.0001 for all agonists). However, 8 weeks of exercise training did not influence the basal platelet reactivity response or EC_50_ to collagen, TRAP6, ADP, U46619, or epinephrine in the presence of DAPT (Figure 2 and Figure 3). An EC_50_ for epinephrine-induced platelet aggregation in the presence of DAPT could not be generated.

### 3.5. Exercise Training and Basal Platelet Prostacyclin Sensitivity

Prostacyclin sensitivity was evaluated as the inhibition of aggregation to a known concentration of platelet agonist as well as the concentration of prostacyclin required to elicit half of the maximal inhibition of platelet aggregation (IC_50_). Maximal platelet aggregation in the presence of vehicle treatment was not different for any agonist pre- versus post-training. Increasing concentrations of prostacyclin resulted in greater inhibition of platelet aggregation (*p* < 0.0001 for all agonists). Prostacyclin sensitivity was improved after training, as evidenced by a ~29% greater inhibition of collagen-induced platelet aggregation (10 µg∙mL^−1^) in the presence of 10 nM prostacyclin (40.9 ± 23.5 vs. 52.9 ± 13.5% inhibition of aggregation; *p* = 0.01) after, compared with before, training (Figure 4A) (no change IC_50_; *p* = 0.27; Figure 5A). Thromboxane A_2_ mimetic U46619 (10 µM)-induced platelet aggregation was inhibited to a greater extent (~99%) after training in the presence of (1 nM) prostacyclin (24.7 ± 18.1 vs. 49.1 ± 11.5% inhibition of aggregation; *p* < 0.0001) compared with before training (Figure 4C) (lower IC_50_; *p* = 0.01; Figure 5E). Lastly, training led to a further inhibition (~44%) of (20 µM) ADP-induced platelet aggregation in the presence of (30 nM) prostacyclin after training (27.5 ± 15.3 vs. 39.6 ± 15.9% inhibition of aggregation; *p* = 0.002) compared with before training (Figure 4D) (lower IC_50_; *p* = 0.001; Figure 5G). Exercise training did not significantly alter the inhibitory effects of prostacyclin on TRAP6-induced platelet aggregation (lower IC_50_; *p* = 0.03; Figure 5C).

### 3.6. Exercise Training and Prostacyclin Sensitivity in the Presence of DAPT

Maximal platelet aggregation in the presence of DAPT treatment was not different for any agonist pre- versus post-training. DAPT elicited a main effect of condition by further facilitating prostacyclin-induced inhibition of collagen-, TRAP6-, ADP-, U46619-, and epinephrine-induced platelet aggregation (*p* < 0.0001 for all agonists). The exercise training program potentiated the inhibitory effects of DAPT in the context of prostacyclin sensitivity; after training, (10 µM) TRAP6-induced platelet aggregation was inhibited to a greater extent (~113%) in the presence of (1 nM) prostacyclin (11.3 ± 7.1 vs. 24.1 ± 15.9% inhibition of aggregation; *p* = 0.01) compared with before training (Figure 4B) (lower IC_50_; *p* = 0.04; Figure 5D). Moreover, the training program resulted in (20 µM) ADP-induced platelet aggregation being further inhibited (~120%) by (1 nM) prostacyclin (6.6 ± 5.0 vs. 14.5 ± 12.2% inhibition of aggregation; *p* = 0.05) compared with before training (Figure 4D) (trend for lower IC_50_; *p* = 0.07; Figure 5H). There was no effect of training on the inhibition of platelet aggregation to collagen (Figure 4A and Figure 5B) or U46619 (Figure 4C and Figure 5F) in the presence of DAPT.

### 3.7. Protein Expression in Isolated Platelets

The exercise training program did not affect the protein expression of thromboxane A_2_ receptor (*p* = 0.83; Figure 6A), thromboxane A_2_ synthase (*p* = 0.67; Figure 6B), cyclooxygenase-1 (*p* = 0.52; Figure 6C), or prostacyclin receptor (*p* = 0.30; Figure 6D) in isolated platelets. Representative blots are shown (Figure 6E).

### 3.8. Plasma 6-Keto PGF_1α_ and Thromboxane B_2_ Levels

Circulating resting 6-keto PGF_1α_ levels were unaltered by 8 weeks of exercise training (73.3 ± 37.5 pg∙mL^−1^) compared with pre-training (71.5 ± 29.1 pg∙mL^−1^) (*p* = 0.90; Figure 7A). Similarly, circulating resting thromboxane B_2_ levels were unchanged after the training intervention (4.62 ± 9.11 ng∙mL^−1^) compared with pre-training (4.74 ± 9.11 ng∙mL^−1^) (*p* = 0.58; Figure 7B). Lastly, there was no difference in the resting plasma 6-keto PGF_1α_ to thromboxane B_2_ ratio before (0.06 ± 0.06 pg∙mL^−1^) and after (0.05 ± 0.03 pg∙mL^−1^) the training intervention (*p* = 0.47).

## 4. Discussion

The primary findings in this study were that 8 weeks of high-intensity exercise training: (1) increased platelet sensitivity to the inhibitory effects of prostacyclin to collagen-, U46619-, and ADP-induced aggregation; (2) potentiated the effect of DAPT on prostacyclin sensitivity to TRAP6- and ADP-induced aggregation; and (3) did not influence platelet protein expression of thromboxane A_2_ receptor, thromboxane A_2_ synthase, cyclooxygenase-1, or prostacyclin receptor.

### 4.1. Exercise Training Improves Basal Prostacyclin Sensitivity

Prostacyclin is a potent inhibitor of platelet aggregation, but only a paucity of studies have examined the role of exercise training on platelet prostacyclin sensitivity and only one study has examined it in postmenopausal women [9]. Our findings are in agreement with previous evidence, while significantly expanding the depth of the literature regarding the underlying mechanisms of action. Similar to Lundberg Slingsby et al. [9], we showed an improved platelet prostacyclin sensitivity to collagen-induced aggregation following 8 weeks of high-intensity training in postmenopausal women (Figure 4A). Interestingly, the participants in the previous investigation were ~3 years postmenopausal, whereas the women in this study were, on average, ~10 years postmenopausal, and a growing body of evidence suggests that cardiovascular adaptations to exercise training may be more difficult to accrue with an increasing number of years after menopause [31]. Accordingly, our novel findings are valuable, because we demonstrate that prostacyclin sensitivity can still be augmented by as little as 8 weeks of exercise training in women that are ~10 years postmenopausal, despite the noted difficulty in obtaining other cardiovascular adaptations. Furthering this point, we expanded on the existing evidence by also showing improvements in platelet prostacyclin sensitivity to TRAP6-, and ADP-induced aggregation after training in postmenopausal women (Figure 4C,D). Moreover, we demonstrate that, after training, lower concentrations of prostacyclin are required to elicit 50% inhibition of maximal platelet aggregation in response to TRAP6 (Figure 5C), U46619 (Figure 5E), and ADP (Figure 5G), providing compelling support that prostacyclin sensitivity is improved via multiple platelet pathways following exercise training. To explore the possible mechanism underpinning the improved sensitivity, we determined the protein expression of the prostacyclin receptor in isolated platelets. Exercise training did not change the receptor expression (Figure 6D), suggesting that the improvement in prostacyclin sensitivity was mediated by an improvement in receptor sensitivity or downstream signaling pathways (e.g., adenylate cyclase or cyclic adenosine monophosphate), rather than receptor quantity.

To understand if training also influenced circulating levels of prostacyclin in the women, we assessed the stable prostacyclin biomarker 6-keto PGF_1α_ before and after the training intervention. In contrast to our initial hypothesis and previous observations [32,33], resting plasma prostacyclin levels were similar before and after training (Figure 7A). However, the previous training intervention showing an increase in systemic plasma prostacyclin levels was in young healthy individuals and 6 months in duration [33]. Therefore, it is plausible that a longer intervention is required to elicit this systemic elevation, particularly in postmenopausal women. In line with this notion, [32] only showed increased local prostacyclin release with acetylcholine infusion with 3 months of training in postmenopausal women.

### 4.2. Exercise Training Potentiates the Effects of DAPT on Prostacyclin Sensitivity

The main goal of this study was to assess the potential synergism between exercise training and DAPT on prostacyclin sensitivity. Previous evidence suggests that the efficacy of DAPT, and specifically P2Y_12_ inhibitors, is highly inter-individualized [34]. However, there is a significant positive association between an individual’s vascular health and the efficacy of DAPT treatment, whereby individuals with greater vascular health are more sensitive to DAPT since P2Y_12_ inhibitors potentiate the anti-platelet effects of prostacyclin and NO [14,15,16]. This led us to hypothesize that exercise training, which has been shown to increase NO [19,20,21] and prostacyclin [32,33] bioavailability as well as prostacyclin sensitivity [9,22], could enhance the inhibitory effects of prostacyclin in the presence of DAPT.

In line with our initial hypothesis, we observed that as little as 8 weeks of exercise training significantly potentiated the response to DAPT in the context of prostacyclin sensitivity. Specifically, we found that after training, the inhibitory effects of DAPT on TRAP-6 and ADP-induced platelet aggregation were enhanced when exposed to low concentrations of prostacyclin, an endogenous platelet inhibitor (Figure 4B,D). Additionally, after training, we observed a significant lowering of the IC_50_ for prostacyclin-induced inhibition of platelet aggregation to TRAP6 (Figure 5D) as well as a trend for a lowering of the IC_50_ for prostacyclin-induced inhibition of platelet aggregation to ADP (Figure 5H). These findings are the first in women but are in line with those of a cross-sectional study in men of differing lifelong physical activity levels [22]. These findings, in conjunction with the previous observation of DAPT (specifically P2Y_12_ receptor antagonists) enhancing prostacyclin sensitivity [35] and our current observation of no change in prostacyclin receptor protein expression (Figure 6D), suggest that exercise training may enhance the efficacy of DAPT on prostacyclin receptor function or downstream signaling pathways (e.g., adenylate cyclase or cyclic adenosine monophosphate).

Taken together, these findings provide novel evidence that postmenopausal women who perform regular exercise training may have greater prostacyclin sensitivity with and without DAPT compared with sedentary postmenopausal women. Accordingly, less DAPT medication may be required for these individuals and personalized dosing schemes could be considered to minimize the problematic side effects of these drugs in terms of bleeding risk [22,23]. Moreover, these data support the notion that regular physical activity is an important tool for promoting anti-thrombotic protection in previously sedentary postmenopausal women.

### 4.3. Exercise Training Increases Basal Platelet Sensitivity to Thromboxane-Induced Aggregation

A scarcity of data exists regarding the effects of exercise training on basal platelet responses in postmenopausal women [9,36]. These authors showed a reduction in platelet reactivity and a concomitant rightward shift in the platelet aggregation curve to several platelet agonists, including thromboxane, in exercise-trained individuals [9,22]. Collectively, these findings reflect reduced platelet sensitivity, and accordingly our contrasting findings were unexpected, as exercise training increased basal platelet aggregation in response to the thromboxane receptor analogue U46619 (Figure 2C,D). Specifically, the increased platelet reactivity to the same concentration of U46619 (1.98 µM) after training may suggest increased sensitivity to this thromboxane receptor analogue. In conjunction with this observation, we report a significant lowering of the EC_50_ (Figure 3E), further demonstrating a sensitization of the platelet aggregation curve to U46619 following 8 weeks of training. Combined, these findings suggest that less thromboxane is required to elicit platelet aggregation following exercise training. However, it is worthwhile to consider that previous findings were in healthy men [22] and premenopausal women [9], and it is possible that the adaptations to exercise training are divergent in postmenopausal women lacking endogenous estrogen. Notably, Lundberg Slingsby et al. [9] reported no effect of 12 weeks of aerobic exercise training on basal platelet reactivity in early postmenopausal women. Although the parameter(s) driving these divergent findings is currently unclear and warrants future investigations, it is possible that menopause and the time after menopause influence the platelet adaptations to exercise training. This may further explain the divergence between our findings in women ~10 years postmenopausal and in the previous study where the women were ~3 years postmenopausal [9]. Moreover, it is important to consider this finding in conjunction with the observed increase in inhibition of U46619-induced aggregation in the presence of prostacyclin following exercise training, where the total in vivo platelet response may favour inhibition of aggregation due to the continuous release of prostacyclin from the endothelium. Nevertheless, to understand the mechanism(s) underpinning the increased sensitivity observed in this study, we explored various components of the thromboxane pathway for platelet activation. However, we did not observe any significant changes in the parameters we measured. Specifically, there were no changes in protein expression of the thromboxane A_2_ receptor in isolated platelets following exercise training, suggesting that increased platelet reactivity was not due to an increase in thromboxane receptor content (Figure 6A). Moreover, we did not observe a change in resting plasma thromboxane B_2_ levels (Figure 7B), or in platelet protein expression of cyclooxygenase-1 or thromboxane A_2_ synthase, enzymes critical for platelet production of thromboxane A_2_ (Figure 6B,C). Interestingly, previous investigations have demonstrated significant reductions in plasma thromboxane B_2_ levels following 16 weeks [33] or 6 months [37] of exercise training, indicating a longer intervention may be necessary for this adaptation.

### 4.4. Exercise Training Does Not Influence Basal Platelet Reactivity with DAPT

Contrary to our initial hypothesis, we did not show any changes in the basal platelet reactivity response to DAPT following the training period (Figure 2 and Figure 3). Our hypothesis was based on the findings from a previous cross-sectional study [22], which showed that lifelong trained middle-aged men present greater basal sensitivity to pharmacological inhibition by DAPT compared with untrained and moderately trained men. However, a lifetime of exercise training elicits numerous cardioprotective effects, making it difficult to compare to 8 weeks of training in previously sedentary postmenopausal women. Accordingly, future studies with longer interventions are required to verify whether exercise training can act synergistically with DAPT on basal platelet reactivity in postmenopausal women. Moreover, this study should be performed in previously sedentary older men to determine the reversibility of inactive aging on platelet function and the responses to DAPT in this cohort.

### 4.5. Study Limitations

DAPT is not a conventional anti-thrombotic therapy for postmenopausal women and is normally only prescribed to prevent further cardiovascular events. Notably, individuals that receive DAPT tend to have additional comorbidities, such as arterial hypertension, diabetes, or obesity. It is, therefore, important to point out that this study was performed in healthy but sedentary postmenopausal women and additional studies are required to elucidate the effects in postmenopausal women at risk of cardiovascular events. DAPT treatment was administered acutely in vitro, which may produce divergent results than in vivo oral administration due to different pharmacokinetic profiles. Furthermore, the women who participated in this study were on average ~ 10 years postmenopausal. Recent evidence suggests that cardiovascular adaptations to exercise training may be more difficult to achieve in this cohort, which may explain our lack of improvement in V̇O_2max_ [31]. Moreover, and perhaps more importantly, a longer training intervention may have produced more dramatic effects on the platelet responses. Accordingly, a greater depth of research is required to determine the optimal training modality and duration for improving platelet function in postmenopausal women as well as whether the number of years after menopause influences the platelet adaptive response to exercise training.

## 5. Conclusions

This study demonstrates that a rather brief period (8 weeks) of high-intensity exercise training is sufficient to induce significant improvements in platelet sensitivity to prostacyclin with and without DAPT in previously sedentary postmenopausal women. This effect may be mediated by improvements in platelet prostacyclin receptor function or downstream signaling pathways (e.g., adenylate cyclase or cyclic adenosine monophosphate), as we did not observe training-induced changes in prostacyclin receptor protein expression or systemic plasma 6-keto PGF_1α_. These findings emphasize the importance of promoting physical activity as a tool for reducing thrombotic risk in postmenopausal women and given the enhanced prostacyclin sensitivity and DAPT enhanced prostacyclin sensitivity after training, healthcare providers may consider training status when prescribing DAPT to this cohort.

## 6. Perspectives

There has been increasing awareness around the underrepresentation of women in pharmaceutical research and the notion that the responses to therapeutic interventions are divergent between men and women [38]. Accordingly, there has been a recent and critical push for the expansion of female inclusion in clinical decision-making to maximize the efficacy and safety of pharmaceutical treatments across the sexes [39]. Notably, this gap extends to DAPT research initiatives [23,24]. The existing literature suggests that after aspirin therapy, women maintain higher platelet reactivity, highlighting a lesser benefit of the treatment [25]. Similar findings have been reported for the P2Y_12_ inhibitor clopidogrel [26]. These data highlight the importance of including postmenopausal women in DAPT research, and this study has taken a step in the right direction for promoting exercise as an additional beneficial therapy for improving platelet function in this cohort. However, the call for inclusion of women in pharmacological and exercise-therapy studies persists and a greater depth of research is required to provide the best individualized care for patients.

## Figures and Tables

**Figure 1 biomolecules-12-01501-f001:**
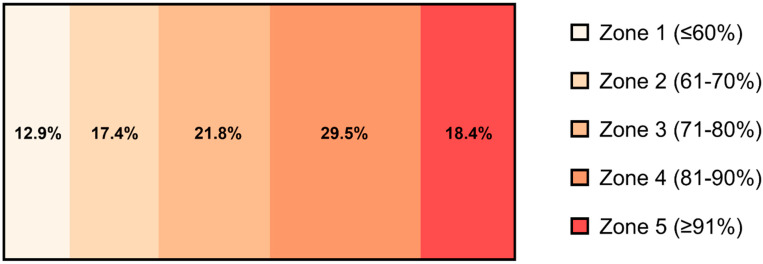
Average time spent in each training zone (%) over the 8-week high-intensity exercise training period (*n* = 13). The training zones were pre-defined by POLAR TEAM Pro. Zone 1 was ≤60% HR_max_, Zone 2 was 61 to 70% HR_max_, Zone 3 was 71 to 80% HR_max_, Zone 4 was 81 to 90% HR_max_, and Zone 5 was ≥91% HR_max_ (Polar Electro Oy, Kempele, Finland).

**Figure 2 biomolecules-12-01501-f002:**
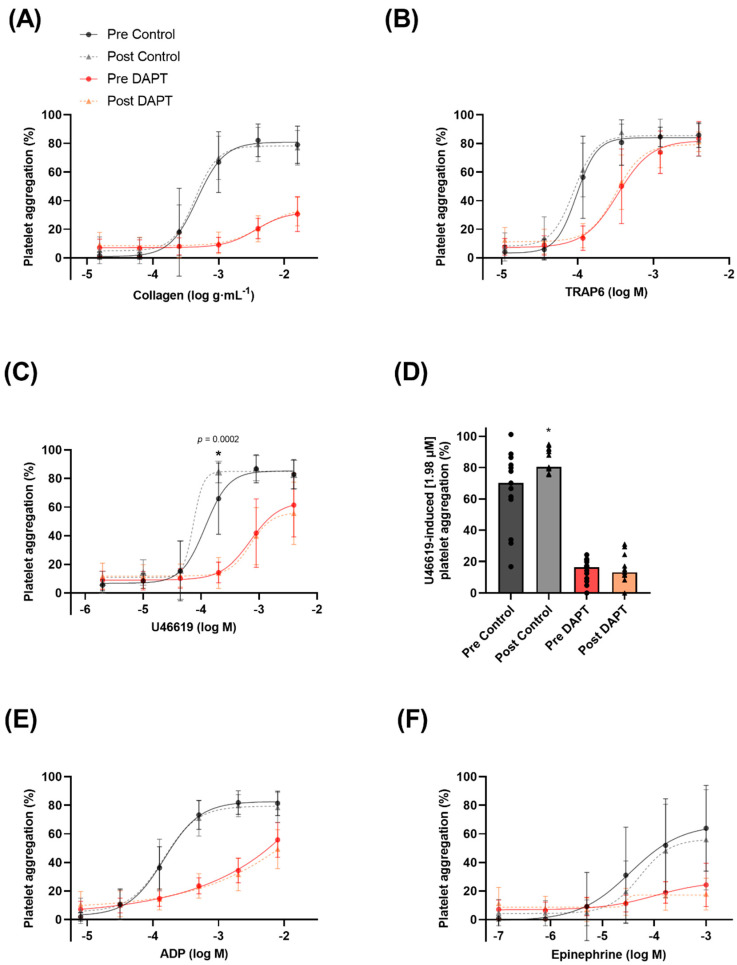
Basal platelet reactivity (expressed as platelet aggregation (%)) to 6 concentrations of the platelet agonists before and after 8 weeks of exercise training with and without dual anti-platelet therapy (DAPT): (**A**) collagen, (**B**) thrombin receptor activator peptide 6 (TRAP6), (**C**) U46619 (U4), (**D**) U46619-induced aggregation at (1.98 µM), (**E**) adenosine 5′-diphosphate (ADP), and (**F**) epinephrine. *, indicates a statistically significant increase compared with pre-training (*n* = 13 for all agonists).

**Figure 3 biomolecules-12-01501-f003:**
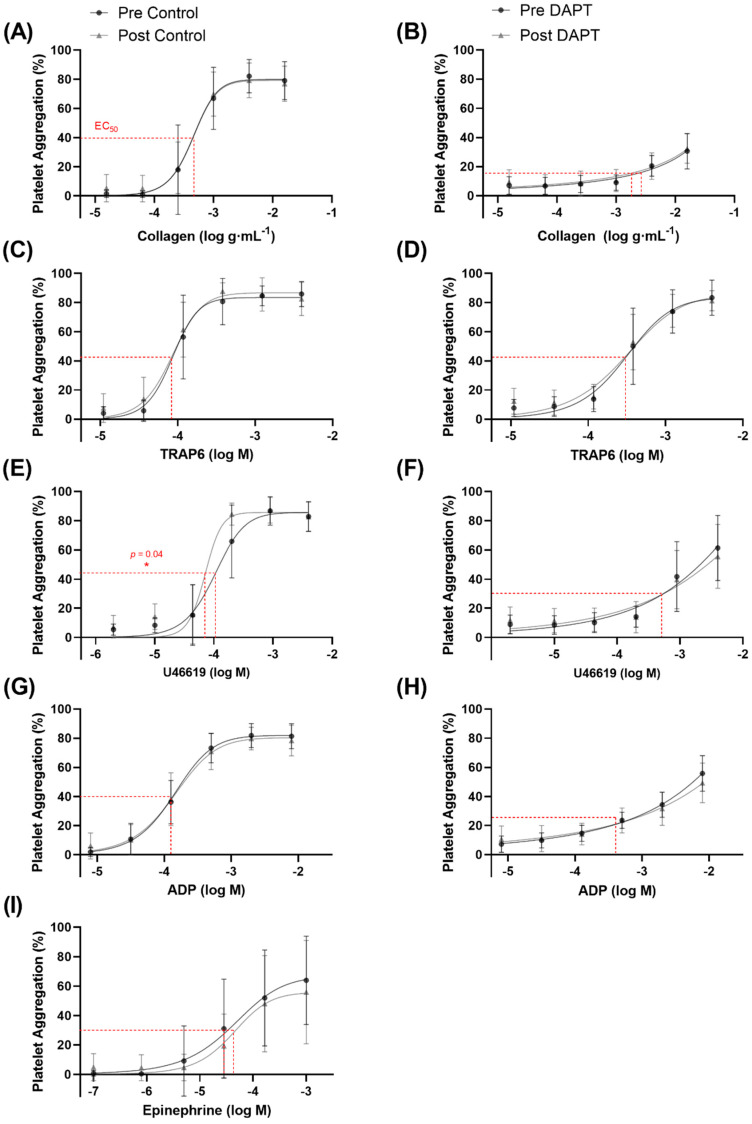
The half maximal effective concentration (EC_50_) for platelet aggregation with (**B**,**D**,**F**,**H**) or without (**A**,**C**,**E**,**G**,**I**) dual anti-platelet therapy (DAPT): (**A**) collagen, (**B**) collagen with DAPT, (**C**) thrombin receptor activator peptide 6 (TRAP6), (**D**) TRAP6 with DAPT, (**E**) U46619, (**F**) U46619 with DAPT, (**G**) ADP, (**H**) ADP with DAPT, and (**I**) epinephrine. *, indicates a statistically significant decrease with training (*n* = 13 for all agonists).

**Figure 4 biomolecules-12-01501-f004:**
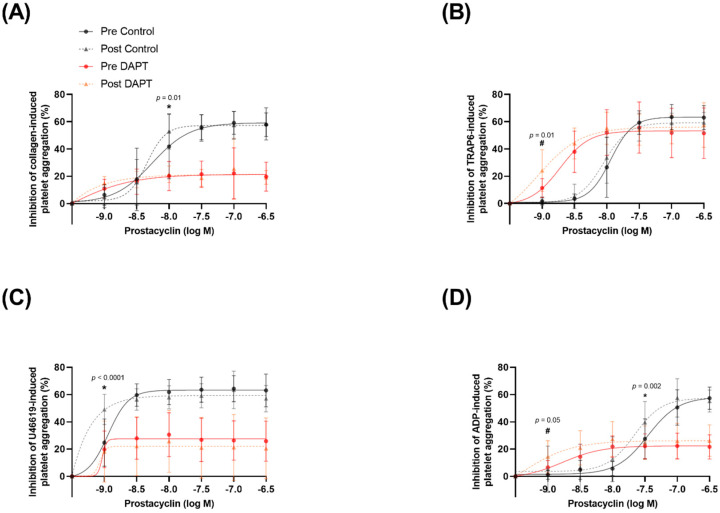
Platelet sensitivity to the anti-aggregatory effects of prostacyclin (1–300 nM), with and without dual anti-platelet therapy (DAPT). Agonist-induced platelet aggregation to (**A**) 10 µg∙mL^−1^ collagen (*n* = 12), (**B**) 10 µM thrombin receptor activator peptide 6 (TRAP6) (*n* = 13), (**C**) 20 µM adenosine 5′-diphosphate (ADP) (*n* = 12), and (**D**) 10 µM U46619 (*n* = 12). *, indicates a statistically significant decrease with training with the control treatment. #, indicates a statistically significant decrease with training with the DAPT treatment.

**Figure 5 biomolecules-12-01501-f005:**
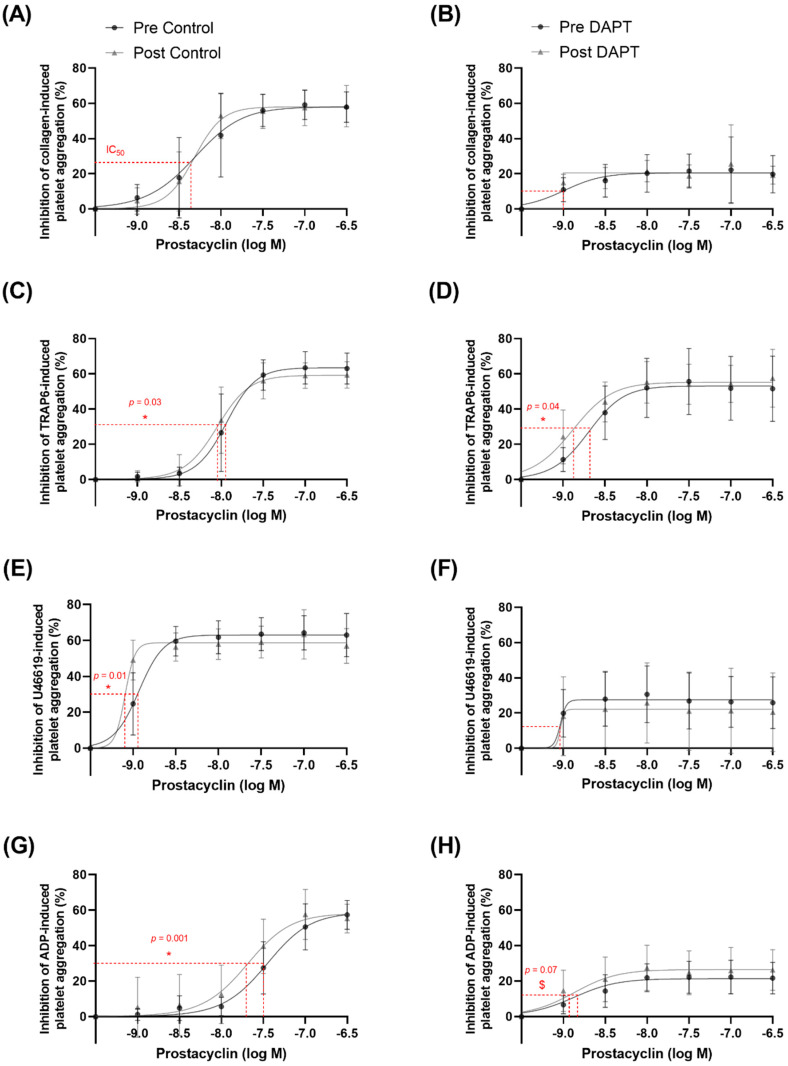
The half maximal inhibitory concentration (IC_50_) for inhibition of platelet aggregation by prostacyclin (1–300 nM) with (**B**,**D**,**F**,**H**) or without (**A**,**C**,**E**,**G**) dual anti-platelet therapy (DAPT). (**A**) collagen (*n* = 12), (**B**) collagen with DAPT (*n* = 12), (**C**) thrombin receptor activator peptide 6 (TRAP6; *n* = 13), (**D**) TRAP6 with DAPT (*n* = 13), (**E**) U46619 (*n* = 12), (**F**) U46619 with DAPT (*n* = 12), (**G**) adenosine 5′-diphosphate (ADP) (*n* = 12), and (**H**) ADP with DAPT (*n* = 12). *, indicates a statistically significant decrease with training. $, indicates a trend for a decrease with training.

**Figure 6 biomolecules-12-01501-f006:**
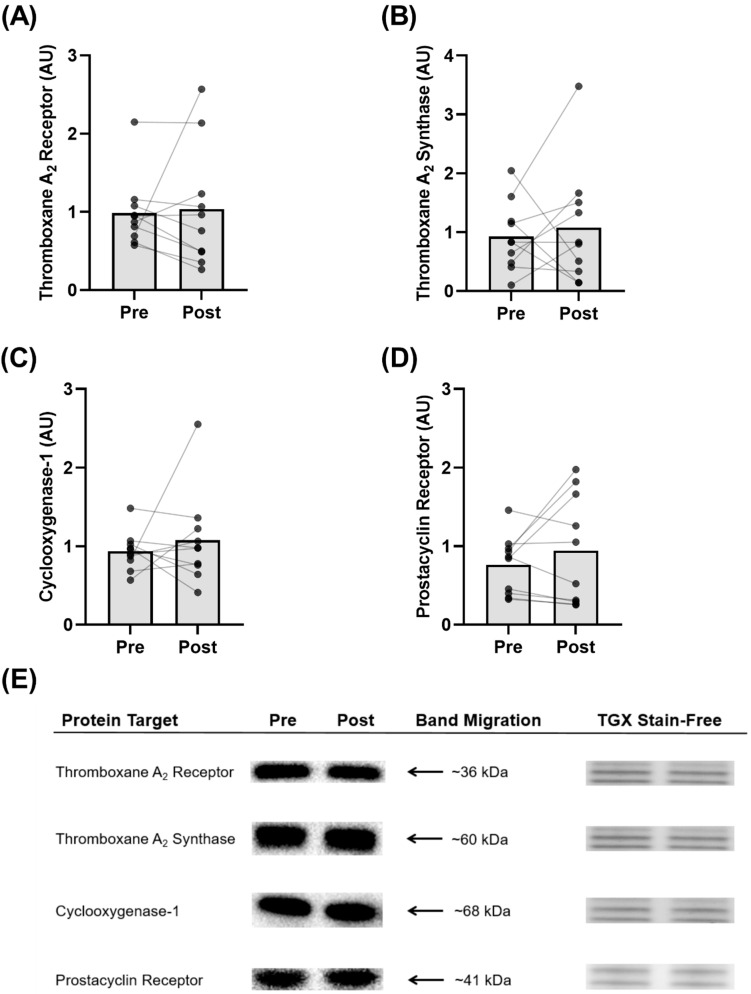
Western blots on platelets isolated from whole blood (*n* = 10 for all proteins): (**A**) Thromboxane A_2_ Synthase (*p* = 0.829), (**B**) Thromboxane A_2_ Receptor (*p* = 0.686), (**C**) Cyclooxygenase-1 (*p* = 0.844), (**D**) Prostacyclin Receptor (*p* = 0.296), and (**E**) Representative Western blots and TGX Stain-Free image. Protein content is expressed in arbitrary units (AU).

**Figure 7 biomolecules-12-01501-f007:**
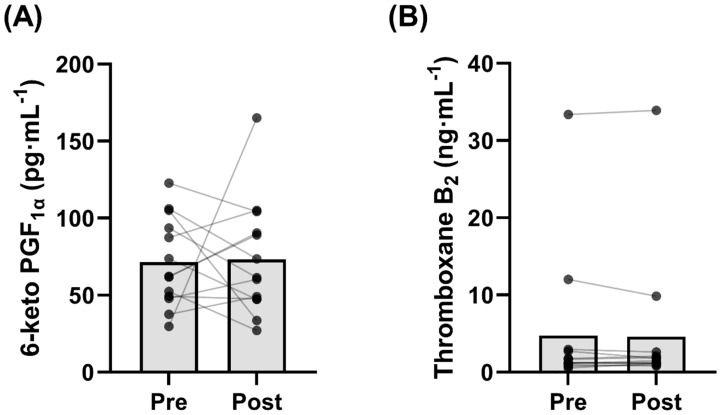
Plasma levels of (**A**) 6-keto prostaglandin F_1α_ (6-keto PGF_1α_) (*p* = 0.895) and (**B**) thromboxane B_2_ (*p* = 0.579) measured pre- and post-training (*n* = 13).

**Table 1 biomolecules-12-01501-t001:** Baseline characteristics of the study participants (*n* = 13).

Participant Characteristics	
Age (years)	61 ± 5
Years After Menopause	10 ± 6
Height (cm)	166 ± 5
Estradiol (nmol·L^−1^)	0.09 ± 0.00
Progesterone (IU·L^−1^)	0.6 ± 0.0
Testosterone (nmol·L^−1^)	0.4 ± 0.1
LH (IU·L^−1^)	36.0 ± 15.2
FSH (IU·L^−1^)	67.2 ± 25.6

**Table 2 biomolecules-12-01501-t002:** Body composition, hematological, and fitness adaptations to 8 weeks of high-intensity exercise training in postmenopausal women (*n* = 13).

	Pre-Training	Post-Training	Statistical Significance
*Body Composition Parameters*			
Body Mass (kg)	72.2 ± 9.0	72.2 ± 8.6	*p* = 0.92
Lean Body Mass (kg)	40.9 ± 3.9	41.5 ± 3.5	*p* = 0.01 *
Fat Mass (kg)	28.8 ± 5.8	28.3 ± 5.7	*p* = 0.16
Lean Mass (%)	60.2 ± 4.4	61.1 ± 4.1	*p* = 0.02 *
Fat Mass (%)	39.8 ± 4.4	38.9 ± 4.1	*p* = 0.02 *
*Hematological Parameters*			
Platelet Count (×10^3^)	180 ± 31	182 ± 28	*p* = 0.76
Red Blood Cells (×10^6^)	5.2 ± 0.9	5.1 ± 1.3	*p* = 0.65
White Blood Cells (×10^6^)	4.2 ± 0.3	4.4 ± 0.3	*p* = 0.06
*Health and Fitness Parameters*			
V̇O_2max_ (mL∙min^−1^)	1895 ± 295	1956 ± 298	*p* = 0.15
V̇O_2max_ (mL∙kg^−1^∙min^−1^)	26.6 ± 5.0	27.4 ± 4.8	*p* = 0.14
Total Cholesterol (mmol·L^−1^)	6.1 ± 0.7	5.7 ± 0.7	*p* = 0.07
Body Mass Index (kg∙m^−2^)	26.0 ± 2.0	26.1 ± 2.0	*p* = 0.74

* indicates a statistically significant change between pre- and post-training.

## Data Availability

The data published in this study is available upon reasonable request and must be in accordance with current General Data Protection Regulation (GDPR) regulations.

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
