# Peer review of "High-Intensity Exercise Training Improves Basal Platelet Prostacyclin Sensitivity and Potentiates the Response to Dual Anti-Platelet Therapy in Postmenopausal Women"

_biomolecules, 2022, doi:10.3390/biom12101501_

Round 1
Reviewer 1 Report
This manuscript raises an important issue and provides relevant results. It is scientifically sound, clear, and presented in a well-structured manner. The study design has been carefully planned. But one small point remains. This experimental study was performed on healthy women. It should be emphasized that DAPT is not a conventional routine antithrombotic therapy in postmenopausal women. There are clear serious indications for its use, such as ACS, stenting, etc. In addition, these women tend to have many comorbidities, such as arterial hypertension, diabetes, obesity, heart failure, etc. These circumstances can affect both the initial coagulation status and its changes against the background of DAPT.
Describing the limitations of their work, the authors rightly point out the discrepancy associated with the use of DAPT in-vitro and in-vivo. Further (p. 19, lines 516-520), they write about the limitation, associated with a cohort of healthy women ~10 years after menopause. However, they forget that DAPT is not prescribed to such healthy women in real life. They did not describe the severe clinical characteristics of DAPT users listed above. Given their influence on coagulation processes, this is significant. Therefore, I recommend expanding the limitations of this study with regard to the clinical status of its participants.
In conclusion, in spite of this minor comment, the article may be recommended for publication.
Author Response
Please see the attached document for our responses to the comments from Reviewer 1. Thank you.

Reviewer 2 Report
Dear Authors,
The menopause, with the subsequent loss of estrogen, is associated with an accelerated rate of endothelial dysfunction, and postmenopausal women have an increased risk of atherothrombotic events compared with premenopausal women. In conclusion, the present study demonstrates that basal platelet reactivity is increased in postmenopausal women, which may contribute to the rapid increase in the risk of cardiovascular events in postmenopausal women. Furthermore, the fact that high-intensity exercise can increase platelet sensitivity to prostacyclin in both pre- and postmenopausal women highlights a previously known mechanism behind the cardioprotective effects of regular exercise. The article is very interesting and well written. The results, figures and tables are clear. I would only recommend rewriting the conclusions with additional details to be more representative of the content of the article.
Sincerely yours,
Author Response
Please see the attached document for our responses to the comments from Reviewer 2. Thank you.
